# Identification of A Novel Arsenic Resistance Transposon Nested in A Mercury Resistance Transposon of *Bacillus* sp. MB24

**DOI:** 10.3390/microorganisms7110566

**Published:** 2019-11-16

**Authors:** Mei-Fang Chien, Ying-Ning Ho, Hui-Erh Yang, Masaru Narita, Keisuke Miyauchi, Ginro Endo, Chieh-Chen Huang

**Affiliations:** 1Graduate School of Environmental Studies, Tohoku University, Aramaki, Aoba-ku, 6-6-20 Aoba, Sendai 980-8579, Japan; meifangchien@tohoku.ac.jp (M.-F.C.); silentboyryan0109@gmail.com (Y.-N.H.); 2Institute of Marine Biology and Center of Excellence for the Oceans, National Taiwan Ocean University, 2 Pei-Ning Road, Keelung 20224, Taiwan; 3Department of Life Sciences, National Chung Hsing University, 145 Xingda Road, Taichung 40227, Taiwan; whereyy@gmail.com; 4Faculty of Engineering, Tohoku Gakuin University, 1-13-1 Chuo, Tagajo, Miyagi 985-8537, Japan; narita-m@tohoku-aep.co.jp (M.N.); kmiya@mail.tohoku-gakuin.ac.jp (K.M.); gendo@mail.tohoku-gakuin.ac.jp (G.E.); 5Tohoku Afforestation and Environmental Protection Ltd., 2-5-1 Honmachi, Aobaku, Sendai 980-0014, Japan

**Keywords:** class II transposon, arsenic resistance operon, mercury resistance operon, genome mining, horizontal gene transfer

## Abstract

A novel Tn*MERI*1-like transposon designated as Tn*MARS*1 was identified from mercury resistant *Bacilli* isolated from Minamata Bay sediment. Two adjacent *ars* operon-like gene clusters, *ars1* and *ars2*, flanked by a pair of 78-bp inverted repeat sequences, which resulted in a 13.8-kbp transposon-like fragment, were found to be sandwiched between two transposable genes of the Tn*MERI*1-like transposon of a mercury resistant bacterium, *Bacillus* sp. MB24. The presence of a single transcription start site in each cluster determined by 5′-RACE suggested that both are operons. Quantitative real time RT-PCR showed that the transcription of the *arsR* genes contained in each operon was induced by arsenite, while *arsR2* responded to arsenite more sensitively and strikingly than *arsR1* did. Further, arsenic resistance complementary experiments showed that the *ars2* operon conferred arsenate and arsenite resistance to an *arsB*-knocked out *Bacillus* host, while the *ars1* operon only raised arsenite resistance slightly. This transposon nested in Tn*MARS*1 was designated as Tn*ARS*1. Multi-gene cluster blast against bacteria and *Bacilli* whole genome sequence databases suggested that Tn*MARS*1 is the first case of a Tn*MERI*1-like transposon combined with an arsenic resistance transposon. The findings of this study suggested that Tn*MERI*1-like transposons could recruit other mobile elements into its genetic structure, and subsequently cause horizontal dissemination of both mercury and arsenic resistances among *Bacilli* in Minamata Bay.

## 1. Introduction

Horizontal gene transfer is a mechanism of giving-and-taking genetic elements between different species and genera, which is a strategy of organisms for survival and plays an important role in microbial evolution. The event of horizontal gene transfer has been confirmed by both laboratory and field studies, and is believed to be driven by environmental stresses, like hazardous pollutions [1]. Mercury and its related compounds are recognized as hazardous pollutants. Among these, the highly toxic methylmercury chloride is widely known as it has been identified as a causative agent of Minamata disease. Though mercury compounds are generally toxic to all living organisms, mercury resistance genes (*mer* operon) were found in many genera of environmental bacteria. Several studies have described the identification and the possible mechanism of horizontal transfer of *mer* genes in both Gram-negative and Gram-positive bacteria [2,3]. The majority of mercury resistance transposons that have been studied are class II transposons, which are characterized by the presence of 35–48-bp terminal inverted repeats (IRs), transposase (*tnpA*) and resolvase (*tnpR*) genes, and a *res*-internal resolution site [4]. *Bacillus megaterium* MB1, an isolate from Minamata Bay, Japan, contains a broad-spectrum mercury resistance determinant encoded in a chromosomal class II transposon, Tn*MERI*1 [4]. Besides *B. megaterium* MB1, 30 *Bacillus* strains were isolated from mercury contaminated Minamata Bay sediment and the genetic characteristics of the *mer* determinants were analyzed [5]. Eleven of the 30 *Bacillus* strains showed broad-spectrum mercury resistance and contained *mer* operons identical to that of *Bacillus megaterium* MB1 [4,6]. However, the localization of these *mer* operons in these *Bacillus* strains remained unclear. On the other hand, with the development of sequencing technology, the whole genome database continues to grow and has proven to be suitable in surveying specific gene clusters. By using genome mining approaches, we have found broad-spectrum mercury resistance gene clusters in not only mercury-polluted Minamata Bay sediment strains but also *Bacillus* sp. strains from water, food and animal sources [7].

On the other hand, arsenic is also recognized as one of the most toxic oxyanions in the natural environment known to cause severe contamination of soil-water systems that has resulted in “blackfoot disease” in endemic areas of some countries such as Bangladesh, India, and Taiwan. Like mercury, many microorganisms have evolved different arsenic detoxification pathways to cope with the widespread distribution of the said pollutant. One of the most characterized pathways is cytoplasmic arsenate reduction and arsenite extrusion [8,9]. Genes for arsenic detoxification were first discovered and characterized in Gram-negative bacteria [8,10]. These genes are often plasmid-encoded and are widespread in prokaryotes [11,12,13,14,15]. In thoroughly studied systems, the *ars* operon has been reported to contain five genes *arsRDABC*, as in *Escherichia coli* [15,16], or at least the three genes *arsRBC*, as in *Staphylococcus aureus* [17]. ArsR is an arsenite-responsive repressor that works in dimeric form and binds to the *ars* promoter [18]. ArsA and ArsB are components of an arsenite-transporter, where ArsA is the ATPase and ArsB is the transmembrane component of the complex. In microorganisms in which ArsA is absent, ArsB acts as a single-component transporter. ArsC is the arsenate reductase that converts arsenate As(V) to arsenite As(III) [19,20,21,22,23], which is subsequently pumped out of the cell through an energy-dependent efflux process [24]. More recently, new arrangements of arsenic-resistance genes have been discovered in a number of microorganisms, including Bacteria [19,20,21,22,23] and Archaea [25]. It is worth noticing that there is a low similarity of 16S rRNA among these microbes, which suggests horizontal transfer among these bacteria. However, there is no report about arsenic resistance mobile elements with mercury resistance ones.

In the present study, Minamata Bay-isolated *Bacillus* containing broad-spectrum mercury resistance were used to investigate the localization of these mercury resistance transposons. A novel type of Tn*MERI*1-like transposon, Tn*MARS*1, was identified, and a transposon-like fragment, namely Tn*ARS*1, was found nested in Tn*MARS*1. The second motivation of this study was to investigate the functions of components in this novel transposon. The results showed that Tn*ARS*1 contains a functional arsenic resistance operon, and that Tn*MARS*1 is a newly isolated Tn*MERI*1-like mercury resistance transposon nesting an arsenic resistance transposon. The diversity of Tn*MERI*1-like transposon in the public *Bacillus* database was also investigated, and the result of genome mining showed that Tn*MARS*1 is the first case in which an arsenic resistance transposon has been identified from a Tn*MERI*1-like transposon. The present study also suggested that the dissemination of Tn*MERI*1-like transposons might contribute to the recruitment of other mobile genetic elements into the *Bacilli* of Minamata Bay.

## 2. Materials and Methods

### 2.1. Bacterial Strains, Plasmids, and Culture Conditions

Eleven mercury resistant *Bacillus* strains (included *Bacillus* sp. MB24) isolated from our previous study were used [5]. *Bacillus* sp. MB24 was used in the 5′-RACE experiment. *Bacillus subtilis* 168 [26] and its derivative with a knocked out *arsB* were used in the arsenic resistance assay. Plasmids pHYARS1, pHYARS2 and pHYARS3 were derivatives of the vector pHY300PLK (Takara Bio Inc., Shiga, Japan), where the *arsR1-orf5*, *arsR2-orf12* and *arsR1-orf12* regions of *Bacillus* sp. MB24 were cloned, respectively, into the *BamH*I and *Xba*I site of pHY300PLK. The bacterial cells were grown in Luria Broth (LB) medium at 37 °C with agitation. Tetracycline and chloramphenicol were added as required at the concentration of 25 μg/mL and 5 μg/mL, respectively.

### 2.2. PCR Amplification of Intact Transposon Regions

The intact mercury resistance transposon regions of the 11 *Bacillus* strains were amplified by using polymerase chain reaction (PCR) with an IR-targeted single PCR primer [4]. The primer sequence is shown in Appendix A, and the conditions of PCR amplification were described in our previous studies [4,27]. The amplicons were digested with restriction endonucleases *Bgl*II, *Hin*dIII, *Nco*I and *Sma*I (Takara Bio Inc.). The digested DNA fragments were electrophoresed on agarose gels and the gels were stained with ethidium bromide (0.5 μg/mL) and subsequently photographed. Restriction fragment length polymorphism (RFLP) profiles of the PCR amplicons were compared with the PCR amplicons of Tn*MERI*1, Tn*5085*, and Tn*5084* as references. The PCR product was analyzed by DNA sequencing.

### 2.3. 5’ Rapid Amplification of cDNA ends (5′ RACE)

The transcription start sites of the *ars* operons were identified by using the 5′-Full RACE Core Set (TaKaRa Bio Inc.). *Bacillus* sp. MB24 was inoculated at 1% (*v*/*v*) in 5 mL of LB broth and incubated at 37 °C with agitation for 3 h. NaAsO_2_ was added to 1 mL of culture to a final concentration of 5 mM, and the incubation was continued for another 15 min. Total RNA of *Bacillus* sp. MB24 was then extracted using the SV Total RNA Isolation System Kit (Promega, Madison, WI, USA) and reverse transcribed through the AMV Reverse transcriptase XL of the 5′-Full RACE Core Set with primers RT-arsR1, RT-ORF3, and RT-arsR2. After removing the RNA by RNaseH, the produced cDNA was purified and self-cyclized by T4 RNA ligase. The cyclized cDNA was subjected to nested PCR with primer sets For-arsR1/Rev-arsR1, For-ORF3/Rev-ORF3, For-arsR2/Rev-arsR2. The primer sequences are shown in Appendix A. The PCR product was analyzed by DNA sequencing.

### 2.4. Real-Time Quantitative Reverse Transcription-PCR (qRT-PCR)

For real time qRT-PCR analysis, *Bacillus* sp. MB24 was inoculated into LB broth and incubated at 37 °C with agitation until the exponential phase. NaAsO_2_ was added to the cultures to a final concentration of 0, 1, 10, 100, and 1000 μM, and the incubation was continued for another 30 min. Total RNA of *Bacillus* sp. MB24 was then extracted using the SV Total RNA Isolation System Kit (Promega) and further treated with DNase I (TaKara Bio Inc.). Real-time qRT-PCR was performed using AccessQuick RT-PCR System (Promega) with SYBR green1 (Bio-Rad, Hercules, CA, USA) along with the Corbett Rotor-Gene 3000 (Qiagen, Hilden, Germany) under the following conditions: reverse transcription at 45 °C for 45 min, followed by 25 amplification cycles, each of which consisted of denaturation for 30 s at 94 °C, annealing for 30 s at 50 °C, extension for 1 min at 72 °C, and finally detecting the fluorescence at 78 °C for 15 s. The primer sets RTarsR1-F/R, RTorf3F/R, and RTarsR2F/R were used to amplify the transcript of *arsR1*, *orf3*, and *arsR2*, respectively. The primer sequences are shown in Appendix A. Primer sets 16F and 16R were used to amplify the bacterial 16S rRNA gene for normalization. Every reaction was repeated at least five times.

### 2.5. Genome Mining of Tn*5084*-like and Tn*ARS*1-like Transposon

A total of 60,922 bacterial completed sequences/loci of Genbank bacterial division (December 2018) and whole genome sequences (4,178 strains) of *Bacillus* (August 2019) from NCBI (National Center for Biotechnology Information) were downloaded, and a selected whole genome sequence database was constructed. The complete sequences of transposon Tn*5084* from *Bacillus cereus* RC607 (GenBank accession number: AB066362.1) and Tn*ARS1* form *Bacillus* sp. MB24 (AY780525) were used as the query sequence against the constructed database for homologous cluster sequence similarity at the level of the entire gene cluster with a 30% sequence identify cut-off and 250 blastp hits mapped per query sequence using the MultiGeneBLAST program [28].

## 3. Results and Discussion

### 3.1. Identification of Tn*MARS*1 in *Bacillus* sp. MB24

Among the 11 broad-spectrum mercury resistant *Bacillus* strains isolated from preserved samples of mercury-polluted Minamata Bay sediment, the mercury resistance determinant of *Bacillus megaterium* MB1 was the most studied [5,7,29]. In our previous study, the mercury resistance determinant, *mer* operon, of *Bacillus* sp. MB24 was identical to that of the MB1 strain [5], while the mercury resistance between MB1 and MB24 strains were different (M.-F. Chien, personal communication). Because the *mer* operon of *B. megaterium* MB1 is located in Tn*MERI*1, a single primer PCR targeting the inverted-repeat (IR) sequence of Tn*MERI*1 was employed to read the intact transposon among these *Bacillus* strains. Three strains, *Bacillus cereus* RC607 having Tn*5084*, *B. megaterium* MB1 having Tn*MERI*1, and *Bacillus* sp. TW6 having Tn*5085*, were used as reference strains. Surprisingly, three types of fragments were detected (Figure 1a). Amplicons from strains MB4, MB5, and MB6 were the same size as the Tn*MERI*1 PCR product (14.2-kbp). Amplicons from strains MB23, MB26, MB27, MB28, and MB29 were the same size as the Tn*5084* and Tn*5085* PCR product (11.5-kbp). However, amplicons from strains MB22, MB24, and MB25 were much longer (more than 25-kbp) than all three transposons (Figure 1a). The amplicon from MB24 strains was sequenced, and the result showed that the *mer* operon and the transposable genes (*tnpA*, *tnpR*) of this amplicon are identical to that of Tn*MERI*1 (M.-F. Chien, personal communication). However, this amplicon lacks the bacterial group II intron located between *tnpR* and *merB3* in Tn*MERI*1. Instead, a 13.8 kbp fragment was nested and resulted in a 25.2 kbp Tn*MERI*1-like transposon. DNA sequencing of this 13.8 kbp fragment showed that it was flanked with a pair of 78-bp IR sequences, which had only one mismatched base. These IR sequences started with GGGG and ended with TAAG, which is common in the Tn*3* family of transposons. At both outer ends of the IR, there were 5 bp of directed repeat (DR) sequences (TAGAA), which are known as the remains of transposition in class II transposons. A total of thirteen open reading frames (ORFs) were identified in this fragment through Blastx analysis in which *rec* and *tnpA* were classified into transposition-related genes, and others were classified into arsenic resistance related genes. This transposon-like fragment was then designated Tn*ARS*1, and the whole 26-kbp transposon in the MB24 strain was designated Tn*MARS*1. The image of Tn*MARS*1 is shown in Figure 1b, in which Tn*ARS*1 was further analyzed.

### 3.2. Characterization of Tn*ARS*1 in *Bacillus* sp. MB24.

#### 3.2.1. Sequence Analysis of Tn*ARS*1

The sequence of Tn*ARS*1 was furthered analyzed. Translated amino acids of *rec* in Tn*ARS*1 showed high similarity to recombinases. The closest one is a recombinase from *Bacillus wiedmannii* (accession no. PTC10770.1), which shares 97% identity. Moreover, the sequence that is needed for resolvase (a kind of recombinase) binding during recombination was also recognized upstream of *rec* in Tn*ARS1*. On the other hand, the translated amino acid sequence of *tnpA* in Tn*ARS*1 showed high similarity to transposases. The closest one is a transposase from *Bacillus stratosphericus* (accession no. KML06687.1), which shares 96% identity. The TnpA in Tn*ARS*1 was classified under the family of transposase Tn*3* based on its conserved domain, DDE motif, which is necessary in catalyzing transposition. These results, including the DR/IR sequence described in *3.1*, suggested an occurrence of a transposition event by Tn*ARS*1, which resulted in the Tn*MARS*1. However, the mismatch of IR, which might have happened during transposition, made Tn*ARS*1 no longer effective as a transposon.

According to the results of Blastp, the arsenic-resistance genes were divided into two gene clusters (*ars* operon) lead by two regulatory genes (*arsR*), *arsR1-orf3-orf4-orf5* and *arsR2-arsB-arsC-arsD-arsA-orf11-orf12*. The *arsR1* and *arsR2* in Tn*ARS*1 were identified as arsenic resistance operon repressor genes. The translated amino acids of *arsR1* and *arsR2* presented a conserved metal-binding motif, EXCVC(D/E)L, and the DNA binding helix-turn-helix motif as with other reported ArsR proteins. The Blastp of ORF3-ORF4-ORF5 did not show similarity with any protein related to arsenic resistance. The conserved domains of ORF3 showed high similarity with glutamate synthase. The conserved domains of ORF4 showed high similarity with a flavoprotein monooxygenase. ORF5 was identified as a GNAT family N-acetyltransferase. The alignment result of the gene cluster *arsR2-arsB-arsC-arsD-arsA-orf11-orf12* showed that the former five genes are typical five-gene type arsenic resistance genes *arsRBCDA*, which are usually found in plasmids of gram-negative bacterium, in which there is no *orf11* and *orf12*. On the other hand, the Blastp results showed similarity to the chromosomal arsenic resistance operon of *Bacillus cereus* Rock1-3, in which ORF11 and ORF12 were also found as a adhesin and protein phosphatase, respectively. The Blastp result of each gene in Tn*ARS*1 is shown in Appendix A. These data suggested that Tn*ARS*1, a 14-kbp unique class II transposon, was inserted into Tn*5085* to form Tn*MARS*1. The sequence of Tn*ARS*1 was deposited into GeneBank with accession number AY780525.

#### 3.2.2. Transcription of *ars* Operons in *Bacillus* sp. MB24

GENETYX-MAC (GENETYX Co.) was employed to predict the potential promoter regions of the two *ars* operons. There were three putative regions: one was upstream *arsR1*, one was upstream *orf3*, and another was upstream *arsR2*. In order to determine the transcription start sites of two *ars* mRNA, total RNA of *Bacillus* sp. MB24 were extracted after inducing by arsenite and were subjected to 5′ RACE. The 5′ terminus of *arsR1* and *arsR2* mRNA were precisely identified to be 30 nt and 97 nt upstream of the initiation codon of ArsR1 and ArsR2, respectively. These results demonstrated that there are two operons, *ars1* operon and *ars2* operon, in Tn*ARS*1. Moreover, no 5′ RACE amplicon was obtained upstream of *orf3*, which suggested a consensus result that *orf3* belongs to the same operon as *arsR1* and thus is transcribed along with *arsR1*. The putative promoter regions (-35, -10) and the transcription start point (+1) of *arsR1* and *arsR2* are shown in Figure 2. The response of *ars1* and *ars2* to different concentrations of arsenic was further investigated by quantitative real time RT-PCR. In the absence of arsenite, transcription of *arsR1* and *arsR2* presented basal levels of 0.1 and 0.06 fold of 16S rRNA transcripts, respectively. The transcription response of the two *arsR* genes responded to different concentrations of arsenite as shown in Figure 3. One micromolar of arsenite was sufficient to induce a more than 4-fold increase of *arsR2* transcription, while the same response of *arsR1* needed 100 μM of arsenite. The transcription induction of both *arsR* genes was positively correlated with the concentration of arsenite. In terms of sensitivity, *arsR2* was observed to increase up to 18-fold of the basal level, while *arsR1* only increased by 4-fold. These results showed that the *arsR2* operon reacts to arsenite more sensitively and strikingly than *arsR1* does. In addition, *arsR1* and *orf3* showed coherent expression profiles suggesting that they belong to the same operon.

#### 3.2.3. Arsenic Resistance Conferred by *ars* Operon in *Bacillus subtilis* 168

The function of the *ars1* and *ars2* operon as an arsenic resistance operon was further investigated. Fragments containing the *ars1* operon, *ars2* operon and both operons were cloned into *Bacillus subtilis* 168Δ*arsB*, which is a strain with a mutated *arsB* in its genome. The minimum inhibition concentration (MIC) and maximum tolerance concentration (MTC) of the transformed strains against arsenite and arsenate were investigated. The results displayed the loss of function of *arsB*, which resulted in 2-fold and 19-fold reductions in the MTCs of arsenite and arsenate, respectively. MIC also exhibited the same trend (Table 1). The reduction was only slightly complemented when introducing the *ars1* operon, but the arsenate and arsenite resistance was significantly recovered by introducing the *ars2* operon into the host bacillus. All of these results revealed that *ars2* operon is a functional operon.

### 3.3. Genome Mining of Tn*5084*-like and Tn*ARS*1-like Regions

There are 11 broad-spectrum mercury resistant *Bacillus* strains isolated from mercury-polluted Minamata Bay sediment [5]. Among these, *Bacillus megaterium* MB1 harbors one group II intron region in the Tn*5084* transposon [30]. In this study, we found that *Bacillus* sp. strains MB4, MB5, MB6 have the same transposon as MB1. Further, we found that *Bacillus* sp. strains MB22, MB24, MB25 also harbor a Tn*5084*-like mercury resistance operon transposon nested by an arsenic resistance transposon (Tn*ARS1*) (Figure 1). It is intriguing that both these two mobile elements are located between the *res* site and the *mer* operon. These results suggested that this region might be a “hot spot” for mobile elements to jump in. In order to survey this phenomenon, we used Tn*5084* as the gene cluster query against a bacterial database (BCT GenBank bacterial division) and the whole genome sequence database of *Bacillus* (4178 whole genome sequences). The results showed that there are some Tn*5084*-like transposons in *Bacillus* sp. strains (Figure 4a). However, there are no transposons or mobile elements nested in the Tn*5084*-like transposon from *Bacillus* sp. except Tn*MERI*1 and Tn*MARS*1. On the other hand, we used the Tn*ARS1* transposon as a query for multi-gene cluster analysis. The results showed that some *Bacillus* sp. strains contain a gene cluster similar to MB24, (*ars1* operon and *ars2* operon), and some *Bacillus* strains harbor only the *ars2* operon (Figure 4b). However, no arsenic operon was identified in a transposon, including any mercury resistance transposon. 

The increase in horizontal gene transfer due to environmental stresses has been demonstrated. Environmental stresses enhance the activity of transposable elements for extensive genomic changes that facilitate the adaptation of populations [31]. The arsenic resistant operon (*ars1* and/or *ars2* operon) were also found in *Bacillus* genomes from different sources, such as rocks, landfill, fermented bacterial culture and patients [32,33,34]. Tn*MERI*1-like transposons were also present in different sources, such as mouse guts, water, beef salad, and canned chocolate beverage [7]. These toxic metal/metalloid resistance operons containing transposons were found in various environments, not only in contaminated sites. However, to the best of our knowledge, Tn*MRS*1 and Tn*MERI*1 are the only samples that an arsenic resistance transposon and a group II intron nested in a mercury resistance transposon, respectively. As they are all identified from Minamata Bay, a site famous for man-induced mercury contamination, anthropogenic pollution might be the main driving force for evolution of the organisms. The new findings of this study might be the result of a horizontal gene transfer event driven by environmental stresses. The presence of Tn*MARS*1 found in Minamata Bay also suggests investigating other mercury contaminated sites, which might reveal further diversity and structure of mercury resistance transposons.

## 4. Conclusions

The results of this study reveal that an arsenic resistance transposon, Tn*ARS*1, was nested in a Tn*5084*-like mercury resistance transposon, designated as Tn*MARS*1. Tn*ARS*1 contains two *ars* operons, in which the *ars2* operon is more functional. Results of genome mining showed that Tn*ARS*1 is the first identified arsenic resistance transposon, and Tn*MARS*1 is the first arsenic resistance transposon nested in another transposon. These results suggest that a genetic element recruitment event has occurred through Tn*MERI*1-like transposons in Minamata Bay, which might be a strategy for *Bacilli* to survive in this severe environment.

## Figures and Tables

**Figure 1 microorganisms-07-00566-f001:**
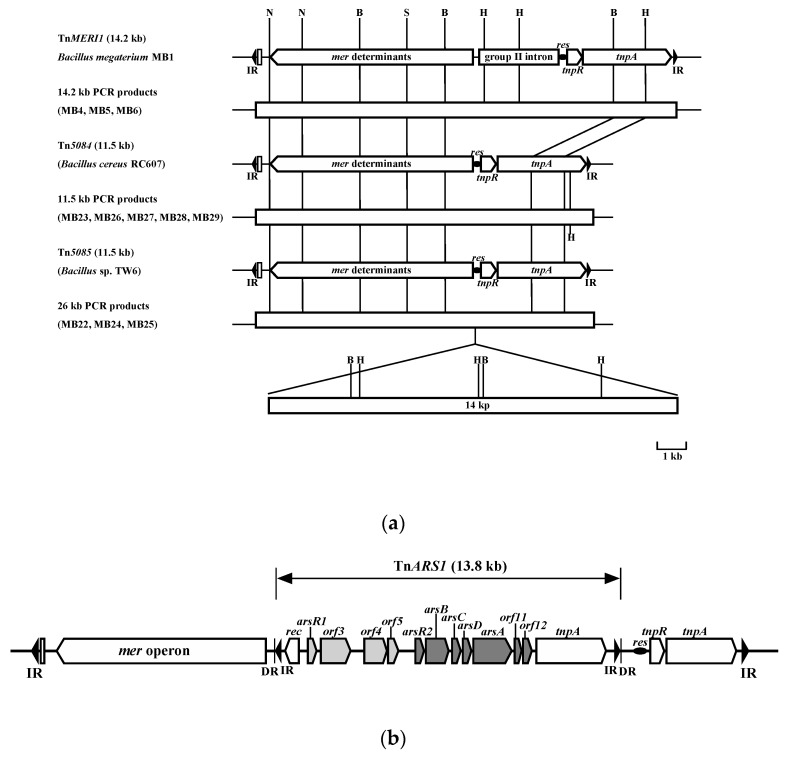
(**a**) The restriction fragment length profile of amplified PCR amplicons from mercury resistant Bacillus strains; (**b**) schematic representation of putative TnMARS1. Hatched boxes indicate the sequenced region.

**Figure 2 microorganisms-07-00566-f002:**
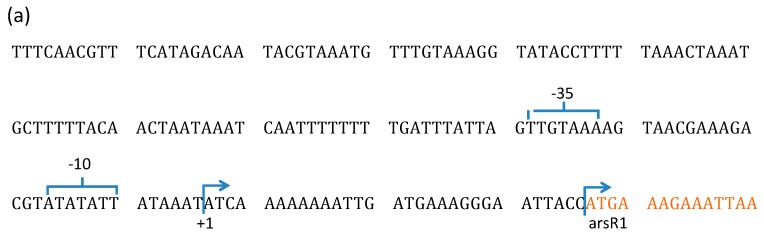
Putative promoter regions (-35, -10) and the transcription start point (+1) of (**a**) *arsR1* and (**b**) *arsR2*. The arrows showed the transcription start sites (+1), and the colored words showed the area of *arsR1* and *arsR2*, respectively.

**Figure 3 microorganisms-07-00566-f003:**
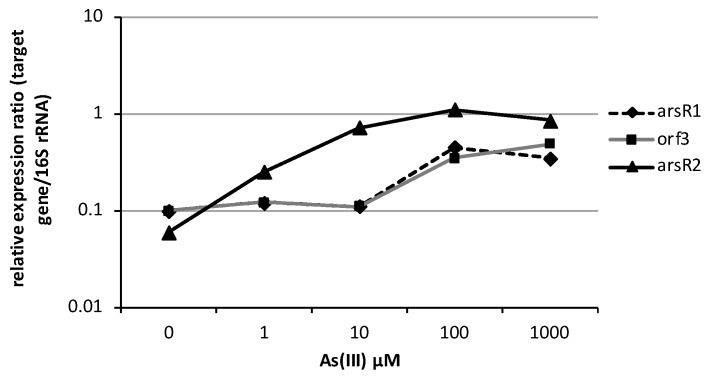
Relative transcription of *ars* operons under different concentrations of arsenite.

**Figure 4 microorganisms-07-00566-f004:**
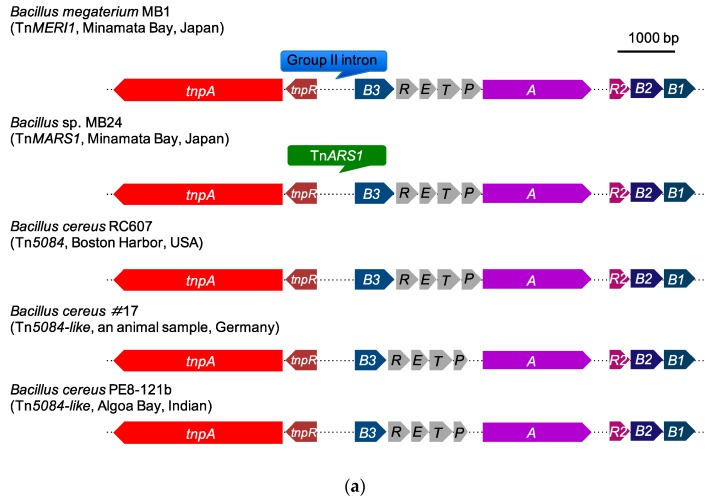
Selected blasting results of (**a**) Tn*5084*-like and (**b**) Tn*ARS*1-like transposons among bacteria by using the MultiGeneBlast analysis. The databases are the BTC database (Genbank: bacterial division, December 2018) and the whole genome sequences of the *Bacillus* database (NCBI, August 2019). The threshold of gene identity is 30% and the gene clusters. *tnpA*: transposase gene; *tnpR*: resolvase gene; *B3*: organomercurial lyase gene *merB3*; *R*: mercury-responsive transcriptional regulator gene *merR*; *E*: mercury resistance gene *merE*; *T*: mercury transpor gene *merT*; *P*: metal binding protein gene *merP*; *A*: Mercuric reductase gene *merA*; *R2*: mercury-responsive transcriptional regulator gene *merR2*; *B2*: organomercurial lyase gene *merB2*; *B1*: organomercurial lyase gene *merB1*. GenBank accession numbers are listed following bacterial strains. Tn*MERI1*: LC152290; Tn*5084*: AB066362.1; *Bacillus cereus* #17: JYFW01000036; *B. cereus* PE8-121b: LRPI01000016.1.

**Table 1 microorganisms-07-00566-t001:** Complementary effect of the ars1 and ars2 operon.

Arsenic Resistance (mM)				
	As(III)		As(V)	
	MTC	MIC	MTC	MIC
*Bacillus subtilis* 168	4	6	38	40
*Bacillus subtilis* 168*ΔarsB*	2	4	2	4
*Bacillus subtilis* 168*ΔarsB*/pHYARS1	2	4	6	8
*Bacillus subtilis* 168*ΔarsB*/pHYARS2	4	6	20	30
*Bacillus subtilis* 168*ΔarsB*/pHYARS3	14	16	10	20
*Bacillus* sp. MB24	12	14	28	30

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
