# Peer review of "Identification of A Novel Arsenic Resistance Transposon Nested in A Mercury Resistance Transposon of Bacillus sp. MB24"

_microorganisms, 2019, doi:10.3390/microorganisms7110566_

Round 1
Reviewer 1 Report
It is a paper that found an arsenic-resistant transposon from mercury-resistant bacteria isolated from Minamata Bay and is worthy of publication.
What kind of compound does the “monooxygense” add oxygen to ?
For example, cyclohexanone monooxygenase or methane monooxygenase, etc.
Author Response
We would like to thank you to kindly and positively evaluate our work.
Regarding to the question about the monooxygenase, we would like to confirm that are you asking the ORF4 in Table S2?
Since this result is a blast result determined by homology, the closest monooxygenase of ORF4 is a predicted flavoprotein CzcO associated with the cation diffusion facilitator CzcD. We have added the flavoprotein monooxygenase to the revised manuscript in Line200 and also Table S2.
Besides that, we have also replied to the reviewer 2’ comments, and the changes had been marked in the revised manuscript. We have carefully revised it and believed that the present manuscript is improved and satisfy the criterion for publication. We appreciate your time and effort to the advanced review.
Reviewer 2 Report
General comment
The manuscript requires extensive language editing.
The biggest worry with the current study is the fact that the authors do not mention why this study was carried out. What was the aim of the study? Why was it important to sequence the whole genome of the isolates? In the last paragraph of the introduction (Lines 78-85), the authors summarise the findings of the present study and do not mention what motivated them to conduct it.
In my opinion, this looks more like the splitting of results that could have been reported in a single study. Thus, a lot of the results presented in the present study are merely an extended discussion of the previous studies (which they cite extensively in the present story). For example, Lines 155-156 (However, amplicons….Fig. 1a) are not different from the observations made in their 2003 paper.
In several instances of the results and discussion, the authors only report their findings and they do not really discuss these findings.
Some minor comments
Line 58: hazardous
Line 61: delete “been”
Author Response
We would like to thank the reviewer 2 for carefully examining our manuscript and point out the concerns for the improvement. We have carefully conducted the revision as shown in the revised manuscript. Please kindly find the changes which are marked in the revised manuscript, and the point by point response is as below.
Besides that, we have also replied to the reviewer 2’ comments, and the changes had been marked in the revised manuscript. We have carefully revised it and believed that the present manuscript is improved and satisfy the criterion for publication. We appreciate your time and effort to the advanced review.
The manuscript requires extensive language editing.
Response 1: Thank you for this comment. We have asked our colleague who is an English-native speaker with PhD degree in environmental molecular microbiology to check the English usage. The minor changes due to English editing were shown in red. We believe the revised manuscript is qualified in English writing.
The aim of this study.Response 2: Thank you for point out this. The aims of this study were using Minamata Bay-isolated Bacillus to investigate: (1) the structures of mercury resistance transposon (2) the functions of components in novel transposon (3) the diversity of TnMERI1-like transposon in public Bacillus database. We have revised the introduction part of the manuscript to make the motivation of this study clearer. We have highlighted the description of motivation in introduction by yellow color. Please kindly find the improvement, and we hope this revised introduction is satisfied.
Why was it important to sequence the whole genome of the isolates?Response 3: The whole genome sequence information could detail the localization of gene clusters and the gene footprints of transposon in bacteria. However, in this case, we didn't sequence the whole genome of these isolates. We applied the reported whole Bacillus genome data base to compare with our new identified transposon to find the possible distribution of TnMERI1-like transposons in bacterial genome pool.
In the last paragraph of the introduction (Lines 78-85), the authors summarise the findings of the present study and do not mention what motivated them to conduct it.Response 3: Thank you. We have added this description of our motivation in the introduction part of revised manuscript. The question which motivated this study is that the localization and construction of the broad-spectrum mercury resistance operon in these Minamata-Bay isolated Bacillus are remain unknown.
In my opinion, this looks more like the splitting of results that could have been reported in a single study. Thus, a lot of the results presented in the present study are merely an extended discussion of the previous studies (which they cite extensively in the present story). For example, Lines 155-156 (However, amplicons….Fig. 1a) are not different from the observations made in their 2003 paper.Response 4: Thank you for evaluating our previous studies. While the present study is to report a novel TnMERI1-like transposon which nests an arsenic resistance transposon. The Fig. 1a of this study showed the structure of whole transposons among the 11 Bacillus, while in our 2003 paper, only parts of the mercury resistance fragments were amplified and shown. In addition, we applied this novel transposon to multigene blast to the database of Bacillus whole genome sequences. This transposon is a new identified and unic TnMERI1-like transposon, which nests an arsenic resistance transposon in Bacillus.
In several instances of the results and discussion, the authors only report their findings and they do not really discuss these findings.Response 5: Thank you for the comments. We have added discussions in the revised manuscript. And the discussion part has been highlighted in yellow color. Please kindly find the changes, and we hope this revision is satisfied.
Some minor commentsLine 58: hazardous
Line 61: delete “been”
Response 6: Thank you for correcting these mistakes. We have corrected them in the revised manuscript.
Round 2
Reviewer 2 Report
The comments have been addressed to satisfaction